# Factors Affecting Shark Detection from Drone Patrols in Southeast Queensland, Eastern Australia

**DOI:** 10.3390/biology11111552

**Published:** 2022-10-23

**Authors:** Jonathan D. Mitchell, Tracey B. Scott-Holland, Paul A. Butcher

**Affiliations:** 1Queensland Government, Department of Agriculture and Fisheries, Animal Science, Ecosciences Precinct, 41 Boggo Road, Dutton Park, QLD 4102, Australia; 2Queensland Government, Department of Agriculture and Fisheries, Fisheries Queensland, 41 George Street, Brisbane, QLD 4000, Australia; 3New South Wales Department of Primary Industries, Fisheries, National Marine Science Centre, Southern Cross University, Coffs Harbour, NSW 2450, Australia

**Keywords:** shark monitoring, beach safety, drones, southeast Queensland

## Abstract

**Simple Summary:**

Drones offer the potential for monitoring sharks at beaches to improve public safety. It is necessary to investigate their effectiveness at detecting sharks in conditions specific to Southeast Queensland, because they have not previously been used in this capacity in this region. The Queensland SharkSmart drone trial ran for 12 months at five beaches, in 2020–2021. The trial ran 3369 flights and sighted 174 sharks (including 48 large sharks over 2 m in length). Sharks were sighted on 3% of flights on average, with North Stradbroke Island having the highest shark sighting rate. We found that location, the sighting of other marine life, season and time of day all had an important impact on the likelihood of sighting a shark from the drones. Overall, we demonstrated that drones could operate across a range of weather conditions and detect sharks effectively. Additionally, the drones provided extra safety benefits because they were used to identify swimmers caught in rip currents and locate missing persons. This research highlights the broad value of drones as a public safety tool at beaches, and the results of this study will help refine the operation of drones to further improve their effectiveness in the future.

**Abstract:**

Drones enable the monitoring for sharks in real-time, enhancing the safety of ocean users with minimal impact on marine life. Yet, the effectiveness of drones for detecting sharks (especially potentially dangerous sharks; i.e., white shark, tiger shark, bull shark) has not yet been tested at Queensland beaches. To determine effectiveness, it is necessary to understand how environmental and operational factors affect the ability of drones to detect sharks. To assess this, we utilised data from the Queensland SharkSmart drone trial, which operated at five southeast Queensland beaches for 12 months in 2020–2021. The trial conducted 3369 flights, covering 1348 km and sighting 174 sharks (48 of which were >2 m in length). Of these, eight bull sharks and one white shark were detected, leading to four beach evacuations. The shark sighting rate was 3% when averaged across all beaches, with North Stradbroke Island (NSI) having the highest sighting rate (17.9%) and Coolum North the lowest (0%). Drone pilots were able to differentiate between key shark species, including white, bull and whaler sharks, and estimate total length of the sharks. Statistical analysis indicated that location, the sighting of other fauna, season and flight number (proxy for time of day) influenced the probability of sighting sharks.

## 1. Introduction

Drones offer a unique perspective of the coastal environment and can provide an effective platform for detecting potentially dangerous sharks in real-time, improving the safety of ocean users. Additionally, drones have minimal impact on the environment or marine life. Drones have been widely used in recent years for marine science research, for quantifying fauna presence [1,2] and behaviour [3,4] to monitoring fishing activity [5,6] and beach usage [7]. Detecting and monitoring sharks from drones to improve public safety is another key area that has recently developed, particularly in Australia [8,9,10]. Drone technology is rapidly advancing, with lightweight, affordable, easy-to-pilot drones now available. As such, drones offer new opportunities to detect sharks in real-time and collect data on the species present and their relative abundance, behaviour and potential risk to ocean users.

Drones have been trialled extensively across New South Wales (NSW) in eastern Australia in recent years and are now operational across >50 beaches throughout the state. Research from this trial has investigated the influence of environmental conditions on the probability of sighting sharks [9,10,11], using advanced camera technology to improve detection [12], the behaviour of white sharks (*Carcharodon carcharias* Linnaeus, 1758) close to surf beaches and near whale carcasses [13,14] and the abundance and diversity of other marine fauna [8,15]. Surveys of public perceptions of drones reported a high level of support (>85%) for their use as a safety tool at beaches, because of their real-time monitoring capability and the fact that they have minimal impact on marine life [16,17]. 

Understanding how environmental variables influence the probability of detecting sharks from drones is important for assessing their effectiveness as a public safety tool. Recent research in NSW and the USA reported that depth and water clarity are the most important variables influencing probability of sighting sharks, although time of day, wind speed and cloud cover can also have an effect [1,9]. When commencing drone monitoring for sharks in new regions, it is necessary to investigate environmental influences, because localised variation may have an important impact on the probability of sighting sharks. For example, prevailing wind conditions, seabed habitat type, depth, turbidity and sea conditions can all change from one beach to the next. In Southeast Queensland, surf beaches have generally high water clarity year round, with a sandy substrate that can enable sharks to be detected due to higher contrast. However, high winds, large waves and heavy rainfall can occur, particularly during summer months. This study therefore sought to assess how an extensive range of environmental variables may influence the probability of sighting sharks across five beaches in Southeast Queensland, over a 12-month period. In particular, the research aimed to (1) identify differences in probability of sighting sharks between survey locations, (2) assess how environmental variables affect probability of sighting sharks, and (3), determine if seasonal variation occurs in relation to 1 and 2. 

## 2. Materials and Methods

### 2.1. Study Location

The Queensland SharkSmart drone trial operated from 19 September 2020 to 4 October 2021, as a partnership between the Queensland Government Department of Agriculture and Fisheries (DAF) and Surf Life Saving Queensland (SLSQ) [18]. Drones were operated at two beaches on the Sunshine Coast (Alexandra Headland and Coolum North), two beaches on the Gold Coast (Southport Main Beach and Burleigh Beach) and one beach on North Stradbroke Island (NSI; Ocean Beach) (Figure 1). These beaches were selected as they are popular locations which have high year-round visitation by a range of ocean users, including swimmers, surfers, kayakers and others and because they are known to have relatively high water clarity. Trial sites were co-located with existing SLSQ services so drone operations could be integrated into existing beach safety operations. Gold Coast city lifeguard services were also involved in drone flight operations at the two Gold Coast beaches. All sites were located outside of CASA regulated airspace that would prevent drone operation, e.g., within 5.5 km of an airport.

### 2.2. Drone Flights

DJI Mavic Pro drones were operated on weekends, public holidays and school holidays by licensed SLSQ pilots, because beach usage by the public was highest on these days and more lifeguards/lifesavers were on duty. Pilots conducted two flights per hour from approximately 8 am until midday. Flights were only conducted in the morning due to prevailing weather conditions in Southeast Queensland, where the wind speeds typically increase in the afternoon. Flights lasted 16.98 ± 3.34 min (mean ± SD) and followed a 400 m transect behind the surf break (Figure 2). The inside edge of the field of view from the transect lined up approximately with the ‘backline’ of the surf break. The position of the surf break can change significantly due to tide and weather variables, so flights were made with manual control (as opposed to automated flight paths). Each flight path extended up to 200 m north and south of the ground control station to stay within visual line of sight, covering up to an 800 m flight circuit (i.e., 200 m north then the drone pivots 180 degrees and heads back 200 m south back to the start point, then 200 m south followed by 200 m north back to the start point). These 400 m transects covered only a portion of each beach close to the lifesaving flags, as opposed to the entirety of the beach, because the pilots were required to always maintain visual line of sight of the drone. Drones were flown at approximately 10–20 km h^−1^ and at a constant altitude of 60 m, providing a field of view width of approximately 110 m with the camera at a 45° angle. The full length of the SLSQ flagged area was included within the flight path. Drones took off and landed from a 30 m exclusion zone on the beach and they were not flown directly above ocean users or people on the beach. To protect privacy of beach users, the drone cameras were only turned on once the drone was above the water.

### 2.3. Data Collection

Drones were set to record continuously in 4K video to maximise the resolution for detecting sharks, and all telemetry data was recorded in the form of accessory. SRT files. All footage was securely archived for later analysis with key operational and environmental data collected for every flight. When a shark was sighted, the drone pilot lowered the aircraft to 10–20 m above the water surface to determine the species [19] and total length while estimating distance of the animal from ocean users. The pilot would then closely follow the shark until it either moved further offshore to the point where it was no longer close to ocean users, or the battery life or distance of the drone from the pilot prevented it. If a shark posed a risk to people in the water, then SLSQ would evacuate people from the water as a safety precaution. Data about the shark sighting was recorded, including the size and species of shark, the time and latitude and longitude as well as the drone height, length of time the shark was followed and its direction of travel. All sightings were verified by the primary author using the recorded footage from the flight. Data for key environmental variables such as wind speed (km h^−1^), sea state (Beaufort scale) and barometric pressure (hPa) were retrieved by the drone pilot, from the Bureau of Meteorology (BOM) website, for each location and every flight. The level of cloud cover in oktas was estimated by the pilot at the start of each flight and the level of turbidity and glare were quantified from video once the drone started the transect. Because they were estimated by the pilots, these three environmental factors were subjective and likely to vary based on different perceptions between pilots.

### 2.4. Data Analysis

Shark sighting rates were calculated at a beach level to identify the percentage of flights where sharks were observed, enabling comparison between beaches. Leopard sharks (*Stegostoma fasciatum*, Hermann, 1783) were excluded from the analyses due to their high abundance at multiple beaches, which would have inflated the number of sharks recorded, and because they pose no risk to ocean users. 

A Generalised Linear Mixed Effects model (GLMM) was applied to determine how a range of environmental and operational factors (Table 1) influenced the probability of detecting sharks. The response variable of the GLMM was modelled with a binomial distribution (presence/absence of sharks). Predictor variables were checked for correlation, which indicated all variable combinations had <0.5 Pearson correlation coefficients. The distribution of predictor variables were also checked and a square root or log +1 transformation was applied to achieve more uniform distributions if necessary. Date in the form of Julian Day from the start of the trial was included as a random factor in the GLMMs to account for any random variation at the day level. To determine the best-fitting model and identify significant variables which explained a meaningful proportion of the deviance in the response variable, we applied a backward stepwise approach to drop individual predictors one step at a time to identify how this changed the Akaike Information Criterion (AIC) values [20]. The best performing model was identified as having the lowest AIC and only those predictor variables which were significant. All statistical analysis was performed in the R language for statistical computing [21]. The package ‘lme4′ was used for running GLMMs [22].

## 3. Results

### 3.1. Operational Results

During the 12 months of drone operations, 3369 individual flights were conducted, covering a distance of 1348 km (Table 2). Drones were able to operate in a range of weather conditions across seasons, although they could not fly in winds greater than 20 km h^−1^ or during rainfall. This resulted in 174 flight days being lost to poor weather across the five beaches combined, which represented 17% of the total number of flight days (Table 2).

### 3.2. Shark Sighting Rates

A total of 174 sharks were sighted across the five beaches. The number of sightings was highly variable across beaches, ranging from no shark sightings at Coolum North to 94 sightings at NSI (Table 3). The majority of these sightings were small whaler sharks (*Carcharhinus* spp.) <2 m in total length, however, 48 large sharks were seen, mostly at Burleigh Beach and NSI. For the three species that may be potentially dangerous to humans (white, tiger (*Galeocerdo cuvier,* Péron & Lesueur, 1822) and bull sharks (*Carcharhinus leucas*, Valenciennes, 1839)), there were two sightings at Burleigh Beach, three at Southport Main Beach and four at NSI, which led to four beach evacuations. No large sharks were sighted at either Alexandra Headland or Coolum North. Drone pilots were generally able to differentiate between the main groups of sharks, including white/bull and whaler sharks (Figure 3). However, in certain ocean conditions such as higher turbidity or if the shark remained close to the seabed, identification to species/group was not possible. Even in optimal water visibility, whaler sharks were not able to be identified to species level, due to their similar morphology. In total, sharks were sighted on 3% of all flights (beaches pooled), with the sighting rates varying from 0% at Coolum North to 17.9% at NSI. Shark sightings occurred on 5.1% of flights at Burleigh Beach, 0.6% at Southport Main Beach and 0.2% at Alexandra Headland. 

### 3.3. Assessing the Influence of Environmental and Operational Factors on the Probability of Sighting a Shark

Drones operated across a wide range of environmental conditions during the trial, providing important data to assess how environmental factors may affect shark sightings. For example, wind speed varied between 0 and 20 km h^−1^ (mean ± SD = 8.2 ± 3.7 km h^−1^) and was recorded from all compass directions, most commonly from the northwest and least often from the west-southwest. Glare and turbidity (which were estimated by the pilot) varied substantially from category 1–5 (mode = 3) and 0– 100% (mean ± SD = 29.6 ± 23.1%), respectively. Other environmental parameters that were likely to influence the sightings of sharks were cloud cover, which ranged from 0–8 oktas (mode = 1) and sea state, spanning 1–6 on the Beaufort scale (mode = 2). Rainfall over the previous 7 days had a large range from 0–548.2 mm, with a mean of 33.5 ± 65.0 mm.

GLMMs indicated that location, the sighting of other fauna, season and flight number were the most important factors that had a significant influence on the probability of sighting sharks, explaining 14% of the deviance in the response variable. The model including these variables had the lowest AIC (Appendix A). The probability of sighting a shark was highest at NSI (0.03), followed by Burleigh Beach (0.009), with Alexandra Headlands and Southport Main Beach having lower values (Figure 4a). No sharks were sighted at Coolum North so this location was not included in the model. The sighting of other fauna increased the likelihood of a shark being sighted (0.009), compared to if other fauna were not sighted (0.004) (Figure 4b). Season also had a variable impact on the probability of shark sightings, with the highest probability of shark sightings occurring in summer (0.03), which was more than double the likelihood for sightings in spring (0.009) (Figure 4c). Sharks were most likely to be sighted on the first two flights of the day (both 0.04) and least likely on flights 3 and 5 (both 0.009) (Figure 4d). There was a low number of occurrences for flight number 8, due to staff availability and weather, so it was excluded from this analysis.

## 4. Discussion

The Queensland SharkSmart drone trial demonstrated the capability for operating drones as a public safety tool to detect sharks, running 3369 flights across five beaches and covering 1348 km. The ability to detect and track sharks at beaches when people were in the water provided a safety benefit because the pilots were able to monitor these sharks in real time and warn ocean users and close the beach if a shark was presenting a threat.

### 4.1. Operational Results

Drones were able to operate across a range of weather conditions, although they were unable to fly in poor weather, which led to 10–23% of days where flights could not be completed. This was expected given the unpredictable and variable weather that occurs on the east coast of Australia, however from a public safety perspective, it is important to note that no people were in the water on the majority of the days when drone flights were cancelled due to the prevailing weather (SLSQ, pers. comm.). Colefax et al. [10] also found that when drone flights were cancelled at NSW beaches due to poor weather, there were no people in the water in 72% of cases. Beaches that were sheltered from certain wind directions, including Alexandra Headlands and Burleigh Beach, had the lowest loss of flights, compared to those that were more exposed, which included NSI and Coolum North. 

From an operational perspective and although not part of the experimental design, it is also important to note that our drones provided a range of public safety benefits at beaches, in addition to detecting sharks. This was demonstrated by their use to identify people caught in rip currents and assist with missing person searches during the trial.

### 4.2. Shark Sighting Rates

Throughout the drone trial, 174 sharks were sighted, 48 of which were large sharks (>2 m total length). Overall, the prevalence of shark sightings was low, with sharks detected on only 3% of flights when all beaches were combined. This result is similar to, albeit slightly higher than, findings from the NSW drone trial, where only 1.9% of flights recorded bull, white and/or whaler sharks [11]. Importantly, there were only nine sightings of bull or white sharks during the current trial, with only four beach evacuations, highlighting that occurrence of these shark species close to beaches are rare, even though they migrate through the study region [26,27,28]. No tiger sharks were sighted during the trial, despite them occurring in this area [26,29] and being caught on drumlines at North Stradbroke Island Ocean Beach during the drone trial period (Queensland Shark Control Program, unpublished data). The lack of tiger shark sightings on drones may have occurred because they typically occur further offshore and thus may be less likely to come in close to beaches and also because they were more active at night, as shown by higher catch data in La Réunion Island [30].

Whaler sharks could not be identified to species level due to their similar morphology, therefore if a whaler shark larger than 2 m total length was sighted by a drone pilot, it was considered to be a potentially dangerous shark and was monitored closely, with the option to evacuate ocean users if it was considered to pose a threat. When reviewing the footage of shark sightings, we found that shovelnose rays (*Aptychotrema* spp.) and white-spotted guitarfish (*Rhynchobatus australiae*, Whitley, 1939) were sometimes mistaken for sharks by the pilots, although not in the cases where beaches were evacuated. This mistaken identity was likely due to their similar silhouette and because they remained close to the seabed, making definitive identification difficult apart from during optimal water visibility. Similarly, both species were commonly misidentified as white sharks during drone operations in NSW (P. Butcher, unpublished data). This highlights the need for pilots to spend time monitoring marine animals to ensure that they are correctly identified and the potential risk to ocean users is mitigated. The use of better camera technology and artificial intelligence [31,32] will aid identification and minimise the number of times beaches are closed unnecessarily.

### 4.3. Assessing the Influence of Environmental and Operational Factors on the Probability of Sighting a Shark

The probability of sighting a shark during drone patrols varied substantially between locations. NSI had the highest probability of shark sightings, and this was possibly due to the very high prevalence of other (non-shark) marine fauna (sighted on 79% of flights at this location), including turtles, rays, large fish and schools of bait fish, all of which can be important prey species for sharks. Indeed, the presence of other fauna was also one of the four main predictor variables influencing the probability of shark sightings in the GLMM analysis, with higher probability when other fauna was present. Colefax et al. [13] and Tucker et al. [14] found that white shark behaviour close to surf beaches was markedly different when food sources were present, with shark swimming speed and track tortuosity (degree of twistedness, i.e., number of turns) increasing, especially in the presence of whale carcasses. The sighting of other fauna may have also had a positive effect in the GLMM because it acted as a proxy for the sightability of sharks, i.e., if the water conditions were clear enough for other fauna to be sighted then they would also enable sharks to be sighted. Interestingly, turbidity did not have a significant influence on sightings of sharks according to the GLMM results, unlike a recent study in NSW, where it had a strong negative impact on sighting rates [9]. This may have been because the turbidity levels were consistently lower at the beaches covered by the current study, as opposed to those in NSW. Additionally, the estimation of turbidity by the pilots was subjective, so the true relationship between turbidity and the probability of sighting sharks may not have been fully captured by the data.

The higher probability of shark sightings at Burleigh Beach was likely influenced by its proximity to Tallebudgera Creek, where outflow brings nutrients into the surrounding area and increases the density of bait fish and other potential shark prey. Higher catches of sharks in Queensland Shark Control Program (SCP) nets and drumlines also occurs at Queensland beaches close to river mouths and after rainfall, especially for bull sharks [33,34]. Higher numbers of fauna sightings were recorded for the beaches closest to river mouths in the NSW drone trial [11], and other research has demonstrated the important link between nutrients and the presence of predators close to river mouths [35]. The very low probability of sighting sharks at the two Sunshine Coast beaches may have occurred because there was less suitable habitat and/or prey species at these locations. The fact sharks were only sighted on 3% of flights overall can act as an important message that sharks are relatively rare at these beaches and the chances of encountering one is minimal. The communication of these facts can improve public knowledge of the likelihood of encountering sharks and increase confidence in ocean users. Such information can also be useful to ocean users on an individual level, when deciding which beach to visit if they are concerned about encountering sharks [36].

Other environmental factors that exerted a significant influence on the probability of shark sightings were season and time of day (flight number). Sharks were more likely to be seen during summer and autumn. Although some species such as white sharks are primarily seen on the east coast of Australia during the austral winter and spring, summer and autumn have higher water temperatures, which lead to greater abundance and activity levels of sharks overall [31,32,37] and higher rainfall, which can lead to greater productivity and prey abundance in the coastal environment due to river outflow carrying nutrients [38,39]. Previous research has also shown that the catch of spinner sharks (*Carcharhinus brevipinna*, Valenciennes, 1839), the most commonly caught species in QSCP nets, was highest in summer months [37]. This species, as well as common blacktip sharks (*Carcharhinus limbatus,* Müller and Henle, 1839), are known to move inshore in Southeast Queensland to breed in spring and summer months [40,41], which would explain the higher numbers of whaler shark sightings we recorded on drones in the current study. However, due to difficulties in identifying whaler sharks from drone footage we cannot be sure that these two species were sighted during the drone trial. Flight number was used as an indicator of time of day, showing that the probability of sighting sharks was greatest on the first two flights of the day, which typically occurred between 7:00 am and 8:30 am. This relationship may have occurred due to higher activity levels of sharks in the early morning [42] and lower levels of disturbance from ocean users and boats in the area at this time. Yet, Ayres et al. [43] reported more frequent sightings of sharks in the afternoon in Cabo Pulmo National Park, possibly linked to thermoregulatory behaviour and tagged white sharks were primarily detected on ocean beaches in eastern Australia in the middle of the day [44]. Wind speed has been found to affect shark sighting rates in other drone-based studies, with lower sightability at higher windspeeds [1,43], however it did not have a significant effect on shark sightability in the current study and the NSW drone trial research also found minimal or no effect of wind on sighting rates [9]. Lastly, Ayres et al. [43] found that increasing temperature and tidal height had a positive relationship with shark sightings from drones. 

It is important to consider that environmental variables can influence both the presence of sharks close to a beach and the ability to detect them from the drone, therefore the latter often limits the ability to determine the former, unless optimal conditions (e.g., clear shallow water <2 m deep in calm conditions) occur during all drone flights. These two processes can interact, for example, recent rainfall can lead to higher outflow from a river mouth, leading to greater productivity and prey abundance for sharks, which would increase the probability of them being present, however the higher turbidity would reduce the likelihood of detecting the sharks from a drone. Therefore, a shark can be present but not detectable, or potentially detectable but not present. Thus, to objectively quantify the effect of environmental variables on sightability, Butcher et al. [9] used shark analogues (model sharks) to assess whether they could be detected under varying environmental conditions, reporting that water depth and turbidity were the most important factors and that detection rates were low when the shark model was at depths >2 m. A study in the Newport River Estuary in North Carolina, USA also used model sharks, finding that shark models were significantly less likely to be detected when water depth was greater than 1 m [1]. Probability of sighting sharks was also lower during the early morning and when cloud cover and wind speed were higher, although these effects were not statistically significant [1]. Hensel et al. [45] found that drones were effective at detecting models of juvenile sharks across a range of benthic substrates (including seagrass meadows, sand and scattered reefs), although depths were <1.5 m. Kelaher et al. [11] reported that water clarity had a significant influence on detectability of a range of marine fauna, whereas sea state and water temperature did not. It is recommended that testing with model sharks is conducted at the beaches surveyed in the current research, with the models deployed at different depths and under varying conditions of glare, turbidity and sea state, to provide a more robust assessment of how environmental variables affect shark sightability.

### 4.4. Future Directions

The technological capabilities of drones are rapidly improving, offering new opportunities to build on the research presented here and optimise the use of drones for detecting sharks in the coastal environment. Artificial Intelligence (AI) is one area which is currently being explored to assess whether it can be used to improve the detection rates of sharks and improve operational efficiency [32,46,47,48]. Recent testing of the SharkSpotter^©^ deep-learning based AI system in NSW, showed that it can achieve a detection success of 90%, which is typically higher than the pilot because they can be distracted by other objects and they are concentrating on flying the drone [47,48]. This AI system operates in near real time and flags any objects detected, which helps to reduce reliance on pilots to observe sharks, improving efficiency and reducing pilot fatigue [48]. Furthermore, Purcell et al., [29] had similar detection rates but also isolated animals into species categories, further enhancing the capabilities and utility for use in beach safety programs. Due to these benefits, AI should be trialled on drones operated at Queensland beaches.

There is potential for advanced camera technologies (e.g., hyper or multispectral cameras) to improve the detection of sharks when conditions are suboptimal, such as when turbidity or glare are higher, by selecting wavelengths that improve water penetration and contrast of sharks [8,49]. Although, the optimal wavelength will likely vary depending on the conditions and locations [8]. Colefax et al. [12] found that the band of wavelengths from 514–554 nm provided optimal contrast between marine fauna (including sharks, dolphins, rays and fish) and the seabed when using a hyperspectral sensor. However, the high cost of hyperspectral sensors, as well as the large amount of data they generate which leads to time consuming post-processing, may limit their cost-efficacy [12]. Alternatively, using spectral filters on a standard RGB camera to select only green wavelengths from 514–554 nm, can achieve similar results to using a hyperspectral camera, but at a much lower cost [12]. These advanced camera technologies and optimal wavelengths should be trialled at Queensland beaches with shark analogues deployed, to determine whether they can improve the detection rate of sharks in conditions specific to Queensland. 

There is scope to expand the deployment of drones to other beaches in Queensland, however before doing so it is prudent to quantify whether each individual beach meets criteria designed to ensure they will be effective. Firstly, environmental conditions need to be considered because some locations (particularly in North Queensland from 12° to 20°) may have high turbidity that makes detection of sharks (if present) unlikely. Indeed, trial flights conducted at Palm Cove near Cairns in North Queensland did not detect any sharks and the water turbidity was often high. Airspace regulations are another key factor that will govern where drones can be operated to detect sharks, because CASA regulations currently prohibit the operation of drones anywhere within a 5.5 km radius of controlled airports. Some beaches in Queensland fall within this regulation, including those on the southern Gold Coast (near Gold Coast airport), mid Sunshine coast (near Maroochydore airport), Cairns and Townsville. There are also limitations on the use of drones in some other areas, such as important bird nesting sites at certain times of year. To maximise the usefulness of the drones, it is advised that they are used at beaches with relatively high year-round visitation rates and which have on-duty lifesavers/lifeguards to operate the drones and with operational processes in place to respond to shark sightings. Another consideration is the historical catch of sharks in SCP gear adjacent to the beach. Those beaches which have a higher catch rate of potentially dangerous sharks due to their biophysical setting (e.g., proximity to an estuary) and/or environmental conditions (e.g., a productive area with lots of baitfish and other potential prey for sharks) should be prioritised as there is a higher likelihood of sharks occurring in these areas. Examples of such locations include Noosa main beach which is close to an estuary and where there is a relatively higher catch of bull sharks compared to other locations, but water clarity is still relatively high. All of these factors influencing the suitability of using drones at different locations have been investigated in a previous report by Cardno [50]. This should be used as an important resource to guide the identification of suitable beaches for drone operations.

## 5. Conclusions

The Queensland SharkSmart drone trial operated for a 12-month period across five southeast Queensland beaches, detecting 174 sharks, demonstrating that drones can operate over a range of conditions and successfully detect sharks close to beaches where water usage is high by the public. Because of this, it is recommended that drones continue to be operated at southeast Queensland beaches, with further expansion to new locations. Opportunities to improve the effectiveness of drones for detecting sharks should also be explored, such as by incorporating AI, optimal wavelengths and beyond visual line of sight flights. The rapid evolution of drone technology will provide significant potential for improving this form of non-lethal shark monitoring into the future and enhancing the protection of ocean users at Queensland beaches.

## Figures and Tables

**Figure 1 biology-11-01552-f001:**
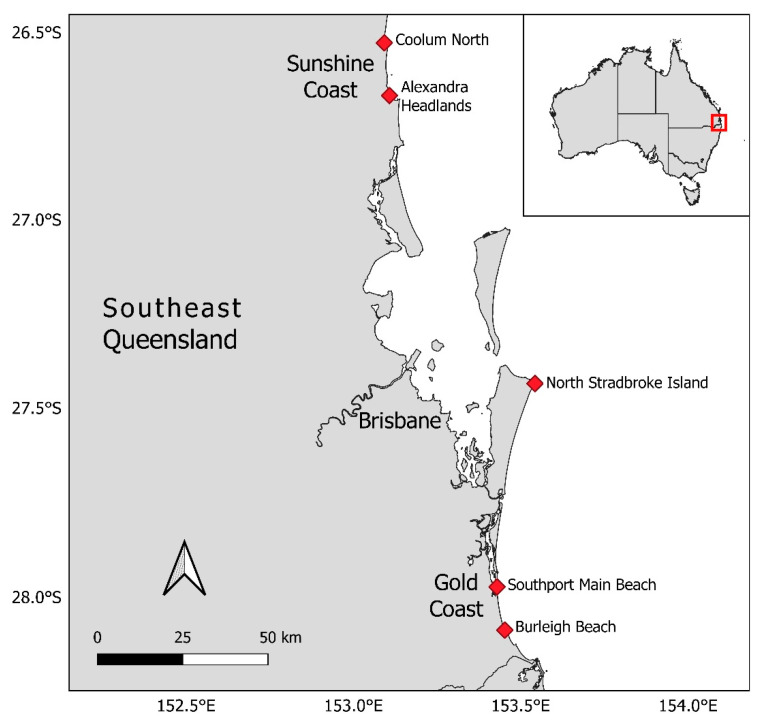
Map of Queensland SharkSmart Drone Trial beach locations in Southeast Queensland, Australia.

**Figure 2 biology-11-01552-f002:**
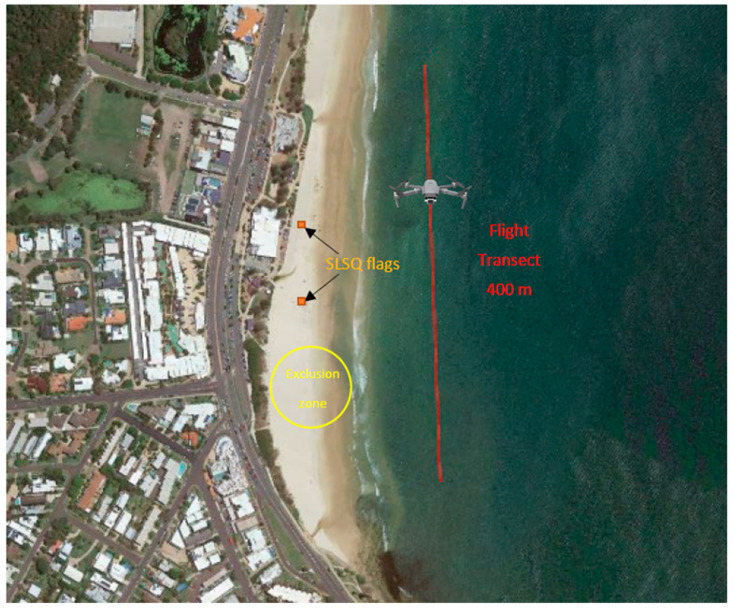
Schematic showing the position of drone transects relative to the flagged area of the beach.

**Figure 3 biology-11-01552-f003:**
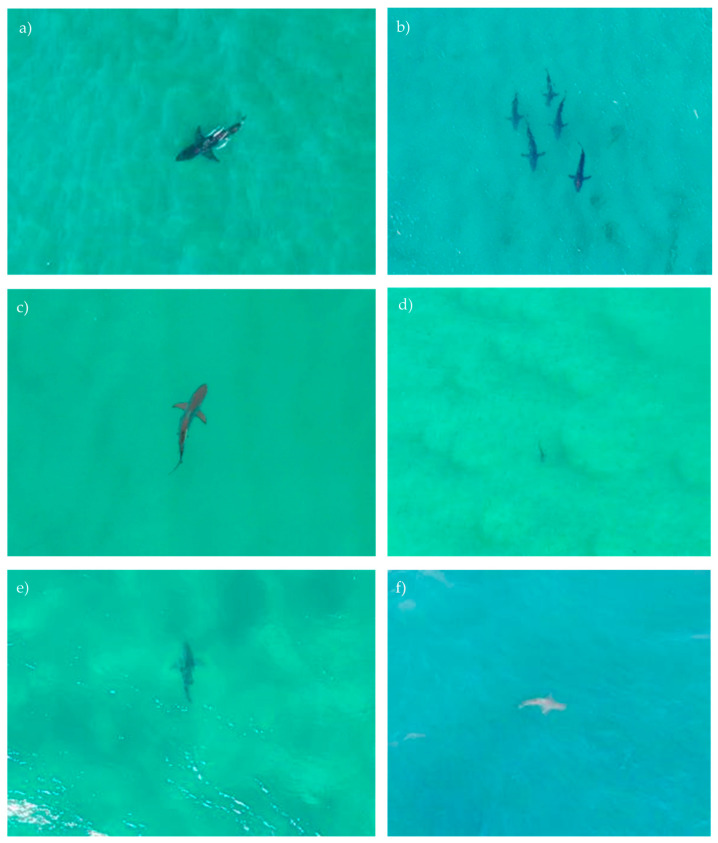
Example images of sharks recorded during the Queensland SharkSmart drone trial. (**a**) white shark (*Carcharodon carcharias*), recorded at Southport Main Beach, Gold Coast in September 2020, (**b**) group of five large whaler sharks observed at Ocean Beach, North Stradbroke Island in November 2020, (**c**) whaler shark (*Carcharhinus* sp.) from the blacktip complex recorded at North Stradbroke Island in December 2020, (**d**) small whaler shark (*Carcharhinus* sp.) seen at North Stradbroke Island in January 2021, (**e**) bull shark (*Carcharhinus leucas*) recorded at Burleigh Beach in June 2021 and (**f**) whaler shark (*Carcharhinus* sp.) at North Stradbroke Island in December 2020.

**Figure 4 biology-11-01552-f004:**
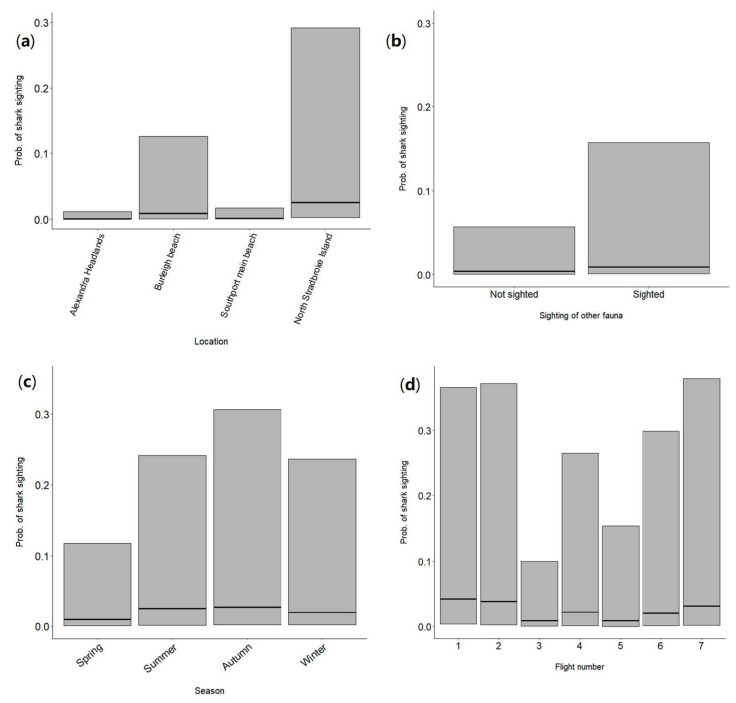
Influence of significant predictor variables on the probability of sighting sharks, across all beaches where sharks were sighted. (**a**) location, (**b**) sighting of other fauna, (**c**) season, (**d**) flight number. Solid black lines indicate model fitted values. Grey shaded areas indicate 95% confidence intervals.

**Table 1 biology-11-01552-t001:** List of environmental and operational variables for which data was collected during the Queensland SharkSmart drone trial, including the metric used and the hypothesized importance of each variable for influencing the probability of sighting sharks.

Variable	Metric Used	Data Source	Hypothesised Importance	Spatial and Temporal Resolution
**Environmental variables**				
Wind speed	Km h^−1^	Bureau of Meteorology (BOM)	Wind speed can lead to increased surface disturbance (e.g., whitecaps) and thus reduce the chance of detecting sharks [1]. Drones were only able to safely operate up to 20 km h^−1^.	From nearest weather station at 30 min intervals. Weather stations were the following distances from beach locations: Alexandra Headland: 6.6 km Coolum North: 6.0 km Burleigh Beach: 11.0 km Southport Main Beach: 1.3 km North Stradbroke Island: 19.0 km
Wind direction	Compass direction	BOM	Wind direction can influence the level of wind disturbance on the water and thus the detectability of sharks	From nearest weather station (see list above) at 30 min intervals
Rainfall	Total rainfall over previous week (mm)	BOM	Rainfall over the previous week can influence the level of turbidity in the water column and therefore the likelihood of sighting sharks	From nearest weather station (see list above)
Cloud cover	Oktas	Estimated by pilot	The level of cloud cover can affect detectability of sharks by influencing the amount of sunlight entering the water and the resulting contrast of sharks against the seabed.	At flight location at start of flight
Barometric pressure	hPa	BOM	Barometric pressure affects weather conditions and can also influence shark behaviour and movements in some cases [23,24]	From nearest weather station (see list above) at 30 min intervals
Sea state	Beaufort Scale (low = 1–high = 12)	BOM	Sea state affects the level of surface disturbance and the ability to see into the water column from a drone	From nearest weather station (see list above) at 30 min intervals
Turbidity	0–100%	Estimated by pilot	Turbidity affects visibility into the water column	At flight location at start of flight
Glare	1 (low)−5 (high) scale	Estimated by pilot	The level of sun glare on the ocean surface can affect the ability of drone pilots to see into the water column	At flight location at start of flight
Presence of other fauna	Presence/absence	Recorded by pilot	Presence of other fauna, especially potential prey species, could attract sharks into the area	Presence or absence of any fauna sighted during the whole flight
Season	Summer-Autumn-Spring-Winter	Recorded by pilot	There are seasonal changes in weather patterns in Southeast Queensland, for example low pressure systems are more common in summer and can cause heavy rain, high winds and rough sea states.	Three month period for each season
**Operational variables**				
Location	Beach	Recorded by pilot	There are differences in habitat type, depth, level of exposure and faunal composition at the five different locations which can influence shark movements and behaviour	Each beach location where flights were conducted
Time of day	Flight number	Recorded by pilot	Time of day affects the angle of the sun and therefore the level of glare and the depth to which sunlight penetrates into the water column. Shark behaviour and movement patterns also vary with time of day [25]	Time that flight occurred. Flight one commenced at 8am and flight eight finished at midday

**Table 2 biology-11-01552-t002:** Operational metrics for each beach covered by the Queensland SharkSmart drone trial. Data covers the time period 19 September 2020–4 October 2021.

Location	Total Number of Flights	Distance Covered (km)	No. of Days Lost to Bad Weather (Percentage of Total Days)
Alexandra Headlands	830	332	20 (10)
Coolum North	759	304	49 (23)
Burleigh Beach	705	282	22 (11)
Southport Main Beach	712	285	34 (16)
North Stradbroke Island	363	145	49 (23)
**TOTAL**	**3369**	**1348**	**174 (17)**

**Table 3 biology-11-01552-t003:** Number of sharks sighted at Queensland SharkSmart drone trial beaches.

Location	Total Number of Sharks ^1^	No. of Large (>2 m) Sharks	No. of White, Bull, Tiger	No. of Beach Evacuations
Alexandra Headlands	3	1	0	0
Coolum North	0	0	0	0
Burleigh Beach	73	23	2	2
Southport Main Beach	4	2	3	0
North Stradbroke Island	94	22	4	2
**TOTAL**	**174**	**48**	**9**	**4**

^1^ total does not include leopard sharks.

## Data Availability

Data for this project are maintained by the Queensland Government, Department of Agriculture and Fisheries, Shark Control Program. Data are available on request by contacting: SCP@daf.qld.gov.au.

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
