# Peer review of "Factors Affecting Shark Detection from Drone Patrols in Southeast Queensland, Eastern Australia"

_biology, 2022, doi:10.3390/biology11111552_

Round 1
Reviewer 1 Report
Summary
This manuscript builds on recent research on the detectability of sharks by drone surveys, taking advantage of field testing done for the Queensland SharkSmart program to evaluate the effect of several environmental variables on shark sightings. The authors set out to assess how an extensive range of environmental variables may influence the detection probability of sharks across five beaches and over a 12 month period. This was addressed using a Generalized Linear Mixed Effects Model, which highlighted several environmental variables that had significant influence on the probability of sighting sharks. The manuscript is clearly written and easy to understand. The study is relevant to the field and particularly to the topic of this Special Issue, expanding the utility of drone surveys conducted for public safety and providing additional information that can be used to manage Human-shark interactions.
General Concept Comments
My major criticism relates to the authors' use of the term detection probability at certain points in the manuscript, most importantly in the title, the abstract, and the aim stated in the Introduction. I believe this term should be used to refer to the probability of detecting a shark given that it is there, which I think the authors do a good job of acknowledging that they can't assess in this study unless conditions are optimal. Therefore, I would suggest that either the authors clearly state they are making an assumption that all sharks that were present were detected or would politely request they rephrase these key sections.
I would also suggest that the authors restructure the conclusions to better reflect the results of the study. I think that there are a couple conclusions that don't reflect the results of this study and I believe that the factors that helped explain the variance in probability of sighting should be mentioned, especially since exploring this was the aim of the study.
Finally, there are a few changes I would suggest to make the methodology a bit more clear. The first is to include a Figure with a map of the beaches surveyed, to help orient readers to the beaches of Southeast Queensland. Secondly, I believe there could be more details included in describing how pilot-estimated variables were obtained, such as turbidity. Finally, I think some reference to the software that statistical analysis was performed in would be helpful.
Specific Comments
Line 2: I suggest dropping the word probability from the title to avoid the confusion I mentioned with what I understand the term detection probability to mean, which I feel is not directly addressed in this study.
Line 39: I suggest changing detection probability to probability of sighting sharks, as it is stated elsewhere in the manuscript.
Lines 78-81: Here as well, I suggest replacing detection probability with probability of sighting sharks.
Line 150: I think there should be some discussion of how pilot-estimated variables were taken, such as at what point(s) during the surveys these were recorded, if they were obtained directly or via drone footage, etc.
Line 170: I think the reader would benefit from knowing what software was used for statistical analysis.
Table 2: I think this would be well complimented with a map of these locations.
Lines 335-339: Does this mean that a "shark" was spotted, but later was not confirmed by 1st author? Also, were there any incidents of evacuations because of one of these mistakes?
Lines 397-399: Since the study does not include temperature or tidal height as variables I don't know that this sentence fits in the discussion, perhaps it could be mentioned as it relates to time of day or otherwise simply mentioned as background information in the Introduction.
Lines 486-488: I'm not sure that this merits being a conclusion of this study since as the authors stated, it wasn't part of the experimental design. I suggest removing this statement.
Lines 488-490: I definitely don't think this sentence belongs in the conclusions, it states the results of another study. Please remove, or better yet, replace with results from the modeling of environmental variables and their influence on the probability of sighting sharks.
Reviewer 2 Report
The authors exemplified with high-quality details and well described results the importance and efficiency of drone technology in shark monitoring in Southeast Queensland, in relation to environmental variables across the study area.
Although the technology employed is not my field of expertise, I truly appreciated the manuscript (easy and pleasant to read) and I followed and agreed on the liaison between the bio-ethology of the species reported and the results presented. Overall, the introduction is concise, but carrying all the necessary information to contextualise the issue. M&M are well detailed, as well as the Results section.
Here is a question, hoping to give a further spark for Discussion (possibly to be integrated in Future directions section): would it be possible to integrate (in real time) the information of sharks' satellite tagging programmes (if available, or to be started ad hoc) with drone recordings? This would be an amazing opportunity to understand how many and which individuals are dwelling in the target sites (especially females of those species known for natal-homing) and help understanding philopatric/migratory behaviour and identifying potential nursery areas to be protected.
As my main focus is molecular taxonomy, here my very minor revisions to the text:
Introduction
Line 60: Please, provide descriptor’s name for Carcharodon carcharias
Lines 110-111: please, be consistent with meters’ formatting across the text. Choose e.g. 800 m or 800m according to the journal’s guidelines
Line 154: Please, provide descriptor’s name for Stegostoma fasciatum
Results
Line 195: Please, provide descriptor’s names for Galeocerdo cuvier and Carcharhinus leucas
Discussion
Line 336: Please, provide descriptor’s name for Rhynchobatus australiae
Reviewer 3 Report
Overall, this manuscript by Mitchell et al., provides insight into how location and environmental conditions affect shark detectability across 5 separate beach locations. While shark detection was low in general, a range of species, including those deemed “dangerous” to humans, were detected. The authors provide reasonable statistical analyses and show that location, presence of other fauna, season, and time of day (flight number) had the most affect on the detectability of sharks within this study. They conclude that these areas are suitable for drone flight operations and shark detection potentially increases beach safety by evacuating beaches when dangerous sharks are detected. However, because of the very low sighting rates and lack of controls, the findings are limited in value and still make efficacy questionable. The authors make conclusions that drones can be used for other aspects of beach safety (missing swimmers, swimmer caught in rip currents), but provide no data to assess the efficacy for these applications. Authors should soften these anecdotal findings or provide data (e.g., number of operations that resulted in aiding lifeguard agencies, number of swimmers helped etc). There are also some ways that data analysis could be improved. For example, Leopard sharks are removed from the study, but it is unclear why as to why. Authors need to be clearer that only potentially “dangerous” sharks are included in the study or include leopard sharks in the analysis. Specifically, if it increases the rate at which sharks are detected this will actually lend evidence to the author’s claims that drones are effective at detecting sharks. Authors could potentially include the presence of conspecifics in their analysis. Simply because a larger group of sharks is more visible than a solitary, potentially missed animal. Most importantly and the authors don’t mention until the very end of the paper is that controls (e.g., shark targets) should have been used to calibrate the effectiveness of the methods being evaluated. Lastly, there are some issues with their figures and tables. Nonetheless, the paper is well written with no fatal flaws within their analyses, and with some revisions this manuscript is recommended for publication.
Specific points are listed below:
Page 2, Line 56-64: If authors would like to make more of a case for drones being used in beach safety other than beach evacuations in the presence of sharks they should include background research that shows that that has been done previously in this paragraph.
Page 3, line 96-98: This sentence is unclear. Perhaps rephrase to be something like, “All sites were located outside of CASA regulated airspace that would prevent drone operation, e.g. within 5.5 km of an airport.” Just something that clears up the statement.
Page 3, line 100-102: Not clear why only on weekends/holidays. Is there a reason these flights were not done during the week? Can be cleared up in a sentence like, “flights were not conducted outside of weekends/holidays/school holidays due to limited lifeguard personnel in the off-season” if that is indeed the case.
Page 3, line 104-110: The description of the survey is unclear. It’s stated that the transect is 400 m in text and the figure, and it’s unclear what/where the ground stations are. Is this for every flight? Then switch to just 800 m in the figure/text. One suggestion is to also describe how much of the beach is actually covered by the surveys. It’s unclear whether these surveys cover the entirety of every beach or only a portion of it. Also, how far offshore do surveys extend? This could affect conclusions if the width of the drone frame is the extent of the surveyed area, as many sharks are farther offshore than 110 m from the wavebreak.
Page 4, line 135: what was the drone’s camera resolution? That’ll affect the area of view.
Page 4, line 139-140: How are shark size and distance estimated? Mavic Pro drones have significant edge distortion which causes overestimation of distances/sizes especially at lower altitudes. Since size is of concern within the paper, is there any quality control for size estimates? There were likely many pilots and a lot of variation between the accuracy of individual pilots.
Page 4, line 148-150 and Table 1: This is all redundant. I would take out Table 1 and insert the reasoning in text or move table 1 to Supplemental. It would also be prudent to add spatial and temporal resolution for all of these variables. Are they measured from buoys that are 10-20 km, or are they modeled data? Over what spatial scale are they modeled? If table 1 is deemed necessary to keep, I’d suggest removing reasoning and add resolutions.
Page 4, 153-154, and page 5 155-156: I think it would have been beneficial to include counting leopard sharks due to their higher abundance, benthic nature, and ease in recognition. It would have provided another calibration for detection of potentially dangerous sharks.
Page 5, line 157-170: As stated, authors should include presence of conspecifics in their models as well, because a higher density of animals and/or shiver of sharks will inherently increase probability of detection as well (see Fig. 2).
Page 9, Table 4: Not sure this Table is necessary. You state ranges and mode in text, so this table doesn’t add anything to the paper. It would benefit the paper to include a table of the best models with AIC values since none of that is stated in the paper.
Page 9, line 260-261: Authors need to rerun analysis with Coolum North included. Just because no sharks were detected does not mean that no sharks were there, thus the entire point of this paper. Taking out Coolum North is biasing the dataset towards areas with higher positive detections. What are the reasons why this was excluded, e.g., sharks have never been sighted at this beach before or detected via drumlines or acoustic receivers. In which case, why survey this location at all?
Page 9, line 267-268: Not clear what “low number of datapoints” means. Is this because weather prevented flight 8 often, or “low number of datapoints” means low number of shark observations. If the latter, this flight needs to be included. See above comment.
Page 10, line 304-305 and page 11, 306-307: Human presence wasn’t measured. Even if it was, there is no indication that fewer people means reduced risk when sharks are observed. It is assumed the beach would still be evacuated if 1 person was in the water or 800, but the number of people in the water would greatly effect encounter rate.
Page 11, line 332-344: It would benefit the paper’s discussion to explore the ecology of the major species observed within the study. Shark observations depend not just on visibility of the actual shark but also whether they are actually present or not. 3% of surveys out of 3,369 surveys is fairly low. Is that because of naturally sparse presence of these sharks in the study sites, or low detectability due to the technology. That isn’t discussed. Lines 363-378 touch on this, but general migration patterns and residency behaviors of these sharks could also explain why sharks were observed or not observed in these locations besides just presence of river mouths.
Page 12, Line 361: so why is that? Were estimates of turbidity inadequate or unreliable?
Round 2
Reviewer 3 Report
While there is still concern about monitoring efficacy without calibrations, the authors have done a better job acknowledging that caveat. They have appropriately toned down conclusions based on study weaknesses.